# Embolization of Recurrent Pulmonary Arteriovenous Malformations by Ethylene Vinyl Alcohol Copolymer (Onyx^®^) in Hereditary Hemorrhagic Telangiectasia: Safety and Efficacy

**DOI:** 10.3390/jpm12071091

**Published:** 2022-06-30

**Authors:** Salim A. Si-Mohamed, Alexandra Cierco, Delphine Gamondes, Lauria Marie Restier, Laura Delagrange, Vincent Cottin, Sophie Dupuis-Girod, Didier Revel

**Affiliations:** 1Department of Cardiovascular and Thoracic Radiology, Louis Pradel Hospital, Hospices Civils de Lyon, 59 Boulevard Pinel, 69500 Bron, France; alexandracierco@gmail.com (A.C.); delphine.gamondes@chu-lyon.fr (D.G.); didier.revel@chu-lyon.fr (D.R.); 2CREATIS, UMR 5220, Univ Lyon, INSA Lyon, Claude Bernard University Lyon 1, 69100 Lyon, France; 3Rockfeller Faculty of Medicine, Lyon Est, Claude Bernard University Lyon 1, 69003 Lyon, France; lauria.restier@etu.univ-lyon1.fr; 4Department of Genetics, Hôpital Femme-Mère-Enfants, Hospices Civils de Lyon, 69677 Bron, France; laura.delagrange@chu-lyon.fr (L.D.); sophie.dupuis-girod@chu-lyon.fr (S.D.-G.); 5Centre National de Référence Pour la Maladie de Rendu-Osler, 69677 Bron, France; 6National Reference Center for Rare Pulmonary Diseases, Louis Pradel Hospital, Hospices Civils de Lyon, UMR 754, INRAE, Claude Bernard University Lyon 1, Member of ERN-LUNG, 69500 Bron, France; vincent.cottin@chu-lyon.fr

**Keywords:** hereditary hemorrhagic telangiectasia, Rendu–Osler–Weber disease, thorax, arteriovenous malformations, embolization

## Abstract

Objectives: To evaluate short- and long-term safety and efficacy of embolization with Onyx^®^ for recurrent pulmonary arteriovenous malformations (PAVMs) in hereditary hemorrhagic telangiectasia (HHT). Methods: In total, 45 consecutive patients (51% women, mean (SD) age 53 (18) years) with HHT referred to a reference center for treatment of recurrent PAVM were retrospectively included from April 2014 to July 2021. Inclusion criteria included evidence of PAVM recurrence on CT or angiography, embolization using Onyx^®^ and a minimal 1-year-follow-up CT or angiography. Success was defined based on the standard of reference criteria on unenhanced CT or pulmonary angiography if a recurrence was suspected. PAVMs were analyzed in consensus by two radiologists. The absence of safety distance, as defined by a too-short distance for coil/plug deployment, i.e., between 0.5 and 1 cm, between the proximal extremity of the primary embolic material used and a healthy upstream artery branch, was reported. Results: In total, 70 PAVM were analyzed. Mean (SD) follow-up was 3 (1.3) years. Safety distance criteria were missing in 33 (47%) PAVMs. All procedures were technically successful, with a short-term occlusion rate of 100% using a mean (SD) of 0.6 (0.5) mL of Onyx^®^. The long-term occlusion rate was 60%. No immediate complication directly related to embolization was reported, nor was any severe long-term complication such as strokes or cerebral abscesses. Conclusions: In HHT, treatment of recurrent PAVM with Onyx^®^ showed satisfactory safety and efficacy, with an immediate occlusion rate of 100% and a long-term rate of 60%.

## 1. Introduction

Arteriovenous malformations are defined as abnormal connections between an artery and a vein and are a common symptom of a rare autosomal dominant orphan disease, the hereditary hemorrhagic telangiectasia (HHT) [1]. Pulmonary arteriovenous malformations (PAVMs) are reported in 30–50% of HHT patients [2]. Because of their abnormal connection, PAVMs bypass the filter of the capillary bed, causing a right-to-left shunt with a high risk of embolic strokes and cerebral abscesses for the patients [3,4,5,6].

Embolization is the standard of care for PAVM treatment [7,8,9]. However, up to 25% of successful embolizations require second treatment due to PAVM recurrence [9,10]. The embolization agents used in the treatment of recurrent PAVMs are solid embolic materials such as coils or plugs. Embolization may be performed according to two different techniques: embolization upstream of the previous embolic materials, more common and technically easier, or embolization downstream of the previous embolic materials, technically difficult but more effective [11]. However, in some cases, retreatment may not be possible because of a too-short afferent artery in case of failure of repeated embolizations (use of numerous coils) or difficult access through the pre-implanted materials [12]. In these cases, the last resort is then surgery.

An ethylene vinyl alcohol copolymer (Onyx^®^), a liquid embolic agent with physico-chemical properties allowing safe and distal embolization, was recently validated in the treatment of cerebral arteriovenous malformations [13]. It was shown to be non-adherent, to have a progressive solidification, a good cohesion, a high vascular penetration and a very weak inflammatory effect on the endothelium [14]. In the lungs, its use for the treatment of naïve treated PAVM is deemed at too high a risk because of its specific architecture showing high flow in the shunt. However, in the case of recurrent PAVM with a lower flow because of the pre-implanted materials, its use may be an appropriate alternative to solid embolic materials and may allow overcoming inaccessible PAVM embolization.

The objectives of this study were to evaluate in the short- and long-term the safety and efficacy of embolization with Onyx^®^ for recurrent pulmonary arteriovenous malformations in HHT.

## 2. Materials and Methods

### 2.1. Study Design

This was a monocentric retrospective study in a reference center. It was approved by the local institutional review board; written consent was waived in accordance with the retrospective character of the study. Clinical, biological and imaging data for all the patients included were extracted from the HHT National Reference Centre database (CIROCO).

### 2.2. Study Population

In total, 62 consecutive patients were retrospectively reviewed from April 2014 to July 2021; 45 patients were eligible for the study, consisting of 70 embolization procedures (flow chart). Inclusion criteria were the following: HHT diagnosis based on the Curaçao criteria [1], evidence of PAVM recurrence on CT and/or angiography, embolization using Onyx^®^ and a 1-year follow-up CT or angiography examination.

### 2.3. Clinical and Biological Data

The standard clinical follow-up consisted of an annual consultation with an HHT specialist at our center, with a pneumologist or with organ specialists when necessary (e.g., hepatologist, cardiologist or neurologist). Clinical and biological parameters were recorded during hospitalization and during follow-up to evaluate the embolization safety and efficacy.

### 2.4. Follow-Up Imaging Protocol

Standard imaging follow-up consisted of unenhanced chest CT one year after embolization. Recurrence was diagnosed based on the standard of reference criteria [8]: efferent vein longer than 2.5 mm and/or increased diameter of the efferent vein or aneurysmal sac. A pulmonary CT angiography or a pulmonary angiography was then performed to conclude the presence of recurrence. This allows defining two groups, i.e., a long-term occlusion (LTO) group for patients with persistent occlusion at follow-up and a short-term occlusion (STO) group for patients with occlusion immediately after embolization but with recurrence at follow-up.

### 2.5. Pulmonary Arteriovenous Malformation Imaging

Two radiologists (with 1 and 6 years of experience) reviewed in consensus, in a random order, all imaging data, i.e., CT before embolization, follow-up CT, pulmonary angiography during embolization and recurrence follow-up pulmonary angiography. In case of difficulty in reaching a consensus, a third radiologist with 15 years of experience was consulted. The radiologists were blinded to patient, PAVMs status and patient’s clinical history for all evaluations. Among the multiple PAVM characteristics collected on CT images, the radiologists recorded the absence of safety distance defined as a too-short distance for coil/plug deployment, i.e., between 0.5 and 1 cm, between the proximal extremity of the primary embolic material used and a healthy upstream artery branch. Recurrence was defined on pre-embolization pulmonary angiogram as recanalization (on the axis perfused by flow through a previously placed coil nest), reperfusion (embolized feeder occluded but presence of small feeders from adjacent normal pulmonary arteries) or both.

Further methods details are provided in Appendix A.

### 2.6. Embolization

Embolization was performed using the routine procedures of our institution, via a common femoral venous access, under local anesthesia. Catheterization was performed using a 5 French catheter through the pulmonary artery and then with a microcatheter to reach a point as distal as possible within the feeding artery so as to deposit the embolic material (Onyx^®^ 18 or coils).

Concerning the embolic material, a dimethyl sulfoxide compatible microcatheter was required to perform supra-selective catheterization of the feeding artery. Onyx^®^ embolization needed flushing of the microcatheter with a saline solution and then with dimethyl sulfoxide to fill the microcatheter’s “dead space”. Onyx^®^ was injected slowly into the feeding artery. It was stopped if a leakage in the upstream arterial branches or in the aneurysmal sac downstream of the embolic materials* was identified on non-subtracted angiography. Immediately after embolization, the efficiency of the treatment was evaluated on a selective non-subtracted angiography to confirm the correct deployment of the embolic material (within, downstream or upstream of the previous embolic material) and on angiography to confirm the complete occlusion (absence of vein opacification) and the pulmonary vascularization in the non-involved arterial territory.

### 2.7. Statistical Analysis

Statistical analyses were performed using the Prism software package (version 8, GraphPad) and the SPSS software (IBM_SPSS Statistics 21; 2020).

All *p*-values < 0.05 were considered significant. Data are expressed as means ± standard deviations (SD) for normally distributed variables and as medians and interquartile ranges (IQR). Categorical variables are described as frequencies and percentages. Differences in diameter for the efferent vein and the aneurysmal sac between pre-embolization and follow-up CT were calculated. Ordinal qualitative variables were compared between the two groups using a non-parametric Mann–Whitney test, and continuous variables using a two-paired Student t-test or a Wilcoxon rank-sum test, and as function of the normality of the variables using the d’Agostino–Pearson test.

## 3. Results

### 3.1. Study Population

In total, 45 consecutive patients (51% women, mean (SD) age 53 (18) years) were retrospectively included from April 2014 to July 2021; 70 embolization procedures were analyzed, corresponding to a mean of 1.4 PAVM per patient (Table 1 and Figure 1). Six PAVMs were treated a second time because of an iterative recurrence after a mean (SD) period of 1.9 (0.7) years. The mean (SD) follow-up period was 3 (1.3) years. Seventeen (24%) PAVMs were treated with a combination of coils and Onyx^®^ because of the poor efficacy of the coils and the absence of distance safety after their deployment.

### 3.2. PAVM Characteristics before Embolization

All PAVMs were initially treated with coils. In total, 55 (86%) PAVMs were simple and similarly distributed in the short- and long-term occlusion groups (STO and LTO groups, respectively) (Table 2). The number of embolizations before Onyx^®^ use was significantly higher in the STO group (2.5 ± 1.3 versus 1.8 ± 1.1 in the LTO group, *p* = 0.01). The vein diameter was significantly higher in the STO group, 5.5 ± 5.6 mm versus 3.4 ± 1.0 mm in the LTO group (*p* < 0.01), as well as the aneurysm diameter, 8 ± 6.1 mm versus 3.4 ± 4.1 mm in the LTO group (*p* < 0.01). Recanalization through the embolic materials was found in 98% of the cases in the LTO and 100% in the STO group. On pulmonary angiography, there was no safety distance in 33 (47%) PAVMs, 17 (60%) PAVMs in the LTO group and 16 (43%) PAVMs in the STO group (Table 2).

### 3.3. Safety

No immediate complication related to the injection of dimethyl sulfoxide and Onyx^®^ was reported. A mild anaphylactic reaction was reported during the pulmonary angiography, which did not require the arrest of the procedure.

No downstream leak in the aneurysmal sac or in the efferent vein or upstream leak in the healthy lobar arteries was reported (Table 3). Upstream leaks in the sub-segmental arteries were reported in 39 (56%) PAVMs and in the segmental arteries in 4 (6%) PAVMs.

No pulmonary perfusion defect was reported in the lobar and segmental lung territories, and 20 (29%) defects were reported in sub-segmental territories. Lung infarctions were reported in three (7%) patients. They resolved spontaneously without requiring longer hospitalization or level 3 analgesics except for one patient for whom it was symptomatic with a 3-day hospitalization. All patients experienced a garlic smell following the dimethyl sulfoxide injection for a couple of days, with no other side effects.

No long-term migration of the Onyx^®^ in the thoracic region was reported, neither brain abscess nor strokes. Hemoptysis due to systemic recruitment of the bronchial arteries from the PAVM was reported in two cases, 3.5 years after retreatment by Onyx^®^ in the first case, related to the systemic reperfusion of a PAVM treated with coils only for the second case. Both were treated by embolization using coils in the bronchial territories.

### 3.4. Short-Term Efficacy

All procedures were technically successful with complete occlusion of the feeding artery. Procedure times were comparable between the LTO and STO groups (110 ± 33 min versus 99 ± 35 min, *p* = 0.15), as well as the volume of Onyx^®^ delivered (0.5 ± 0.3 in the STO versus 0.7 ± 0.6 mL in the LTO group, *p* = 0.23) (Table 3). Onyx^®^ filled the inside of the previously delivered coils in 73% of cases and the upstream artery in 69% of cases. Case examples are provided in Figure 2, Figure 3, Figure 4, Figure 5 and Figure 6.

### 3.5. Long-Term Efficacy

Recurrence was suspected on CT in 27 patients (60%), a total of 42 PAVMs. Recurrence was further evaluated with pulmonary angiography for 30 PAVMs (25 examinations, 71%), among which 13 (43%) did not show evidence of recurrence, or with pulmonary CT angiography for 12 PAVMs (29%), among which one (8%) did not report recurrence evidence. Overall, 14 (47%) PAVMs showed no evidence of recurrence despite a non-reduction of the vein or aneurysm diameter (Table 4).

A persistent occlusion with a mean reduction between before and after embolization of the aneurysm and vein diameter of 40% and 30% were reported in 42 (60%) PAVMs. On CT follow-up, the vein was significantly larger in the LTO group (3.8 ± 1.0 mm diameter versus 2.4 ± 0.9 mm in the STO group, *p* < 0.001) as well as the aneurysm (7.0 ± 5.1 mm versus 2.1 ± 3.2 mm, respectively, *p* < 0.001). To note, the after rate of success was 56.2% in PAVM treated only once with Onyx^®^.

## 4. Discussion

In this retrospective study conducted in an expert center, the safety and efficacy of the Onyx^®^ liquid embolic agent were demonstrated for the embolization of recurrent pulmonary arteriovenous malformation in a hereditary hemorrhagic telangiectasia population. This technique allowed distal endovascular embolization, particularly for PAVMs not eligible for additional coils or plug embolization because of a high risk of occlusion of the collateral branch. All procedures were technically successful, with an immediate occlusion rate of 100%. In the long term, a 60% occlusion rate was reported, with no complications related to the embolization procedure.

Treatment of recurrent PAVM is a challenge. The success rates found in the literature vary from 0 to 80% [11,12,15,16]. In the present study, the long-term occlusion rate was 60%, with a specific 58% rate for recanalized PAVMs and 67% for both recanalized and reperfused PAVMs, in accordance with previous studies. Woodward et al. showed a 66% occlusion rate in 38 PAVMs and 83% for recanalized PAVMs [10]. Another team, Milic et al., showed a 42% occlusion rate in 33 PAVMs (19 patients) [12]. Additionally, embolization failed in 20% of the cases due to the absence of distance safety.

Some baseline variables were significantly different in the success and failure groups and may be determinant factors to consider before embolization. Recurrent PAVMs of the failure group presented the highest number of embolotherapy before retreatment with Onyx^®^, which may indicate a complicated and refractory type of PAVMs. They also presented large veins and aneurysms, which was not found in a previous study that showed that smaller PAVMs were associated with a higher rate of reperfusion [17]. The presence of a large feeding artery was shown to be a factor of recurrence [12] but was not evaluated in this study because of the pre-implanted embolic materials. Last, the proportion of PAVMs with coils deployed at more than 10 mm from the aneurysm, considered a factor of recurrence [12], was slightly higher in the STO group. Altogether, in our study, the recurrent PAVMs treated with Onyx^®^ were comparable to those treated with standard embolic materials in previous studies.

This study reports, to our knowledge, for the first time, results of embolization of recurrent PAVMs using Onyx^®^. The choice of this embolic material was supported by the need to fill the pre-implanted materials, as shown in more than 70% of cases in which Onyx^®^ filled the coils packing. Contrary to the glue, Onyx^®^ does not present adhesive properties when in contact with the arterial walls but has “filling” properties which may have facilitated its use for slow and controlled distribution around the pre-implanted materials. It was supported by the lack of a safety distance between the pre-implanted materials and healthy collateral, as reported in 47% of the PAVMs treated. The criteria for embolization arrest were defined before the study started in consensus by our team, based on previous data of Onyx^®^ embolization in other locations and on our experience in limiting the risk of a leak in the systemic circulation and in healthy pulmonary territories. Despite the evidence for treating the nidus in addition to the feeding artery in PAVM naïve of embolization [18,19], we avoided downstream leakages by stopping the procedure when Onyx^®^ would go past the materials, which occurred in 19% of cases. However, no further leak was reported neither in the aneurysm nor in the efferent vein or in the systemic circulation, as confirmed during angiography or follow-up chest CT. The procedure was also stopped when Onyx^®^ would go upstream of the pre-implanted embolic material in a healthy arterial branch. Nevertheless, in 69% of cases, an upstream leak in a non-involved arterial branch was reported, which opens to injection techniques under flow control [20]. Despite this high proportion, only 29% of these cases presented a perfusion defect on pulmonary angiography, from which only 4% of the patients reported a distal lung infarction, which was quasi-asymptomatic and resolved spontaneously. This low rate of perfusion defect, compared to the number of leaks in collaterals, may be explained by the non-obstructive deposition of Onyx^®^ within the healthy artery, hardly differentiable from an obstructive deposition due to the opacity of this material on pulmonary angiogram. In addition, the low rate of lung infarctions, compared to the number of perfusion defects, may be explained either by the presence of asymptomatic infarctions or secondary recurrences of the embolized territory. Nevertheless, this complication is well-known and frequently reported in the endovascular treatment of PAVM [7] and would probably have been more frequent using coils or plugs because of the absence of a significant safety distance [21]. Finally, follow-up of some PAVMs showed no reduction in vein diameter or aneurysm size despite persistent occlusion, which raises the question of the expected reduction in PAVM size [8]. In our practice, we hypothesized this by a loss in vascular compliance after iterative embolization, opening to furthermore investigations.

According to our experience, the success of the procedure was defined according to the standard of reference, i.e., the absence of vein opacification during pulmonary angiography [8]. All procedures resulted in a complete occlusion immediately after Onyx^®^ injection. The mean injected volume was 0.6 mL, low compared to that injected for cutaneous or cerebral arteriovenous malformations [13]. This may be explained by the specific angioarchitecture of PAVMs with a limited volume spare of a recanalized feeding artery and the absence of a nidus. That may explain that contrary to certain techniques of embolization with Onyx^®^ that require a waiting time for polymerization before a second injection, we injected Onyx^®^ continuously until the endpoint was reached.

This study has some limitations, mainly its retrospective and monocentric character. Additionally, the lack of a reference method for the diagnosis of persistent occlusion, i.e., pulmonary angiography, is a limitation. This choice was based on both the current practice in our expert center and on previous results showing sensitivity for recurrence of 98.4% for PAVMs with a vein diameter larger than 2.5 mm (10). Nevertheless, in case of a vein diameter higher than 2.5 mm and/or a recurrent diameter of the aneurysm sac and vein, we performed an injected examination in order to confirm the recurrence.

## 5. Conclusions

Embolization with Onyx^®^ of recurrent pulmonary arteriovenous malformations allowed a short-term occlusion rate of 100% and a long-term rate of 60%, offering an additional option for the treatment of challenging recurrent pulmonary arteriovenous malformations in HHT.

## Figures and Tables

**Figure 1 jpm-12-01091-f001:**
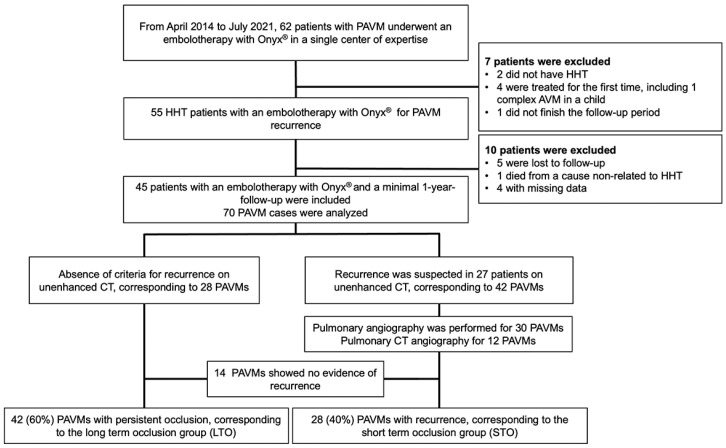
Study flowchart.

**Figure 2 jpm-12-01091-f002:**
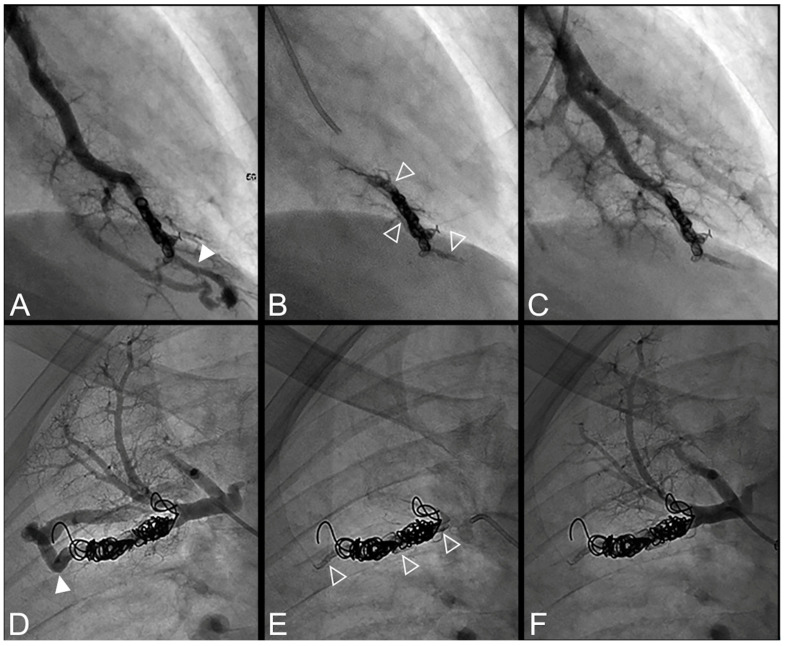
Case examples of a 68-year-old man (**A**–**C**) and 48-year-old man (**D**–**F**) treated for a simple recurrent pulmonary arteriovenous malformation. In both cases, digital subtraction angiography unsubtracted images showed a distance >10 mm between the first coil and the aneurysmal sac, which is considered a risk factor for recanalization. (**A**–**C**). Embolization was performed using Onyx^®^ (0.3 mL) to fill the afferent artery in and downstream of the pre-implanted coils and resulted in an immediate complete occlusion, maintained after 23 months follow-up. No leak in the aneurysm or in the vein was reported. (**A**). Opacification of the afferent artery showed a recanalization through the pre-implanted coils (full arrowhead). (**B**). Opacity within, downstream and upstream of the coils (empty arrowheads) showed the distribution of Onyx^®^ without any evidence of a leak in the aneurysmal sac. (**C**). Opacification of the afferent artery showed the absence of opacification of the aneurysmal sac and the efferent vein in favor of immediate occlusion. The opacification of the healthy arterial branch did not reveal any perfusion defect. (**D**–**F**). Embolization was performed using Onyx^®^ (0.4 mL) to fill the afferent artery in and downstream of the pre-implanted coils and resulted in an immediate complete occlusion, with a recurrence 36 months after the procedure. No leak in the aneurysm or in the vein was reported. (**D**). Opacification of the afferent artery showed a recanalization through the pre-implanted coils (full arrowhead). (**E**). Opacity within, downstream and upstream of the coils (empty arrowheads) showed the distribution of Onyx^®^ without any evidence of a leak in the aneurysmal sac. (**F**). Opacification of the afferent artery showed the absence of opacification of the aneurysmal sac and the efferent vein in favor of immediate occlusion.

**Figure 3 jpm-12-01091-f003:**
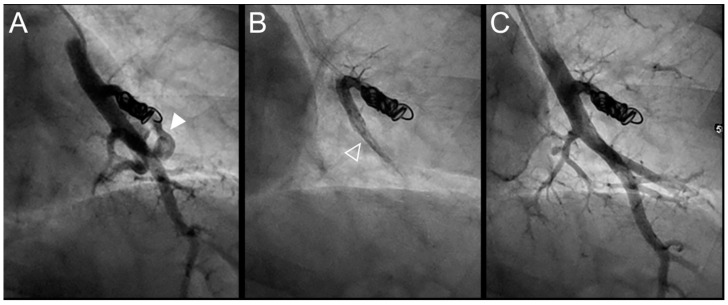
Case example of a 57-year-old woman treated for a simple recurrent pulmonary arteriovenous malformation (PAVM) in the lower left lobe. Digital subtraction angiography unsubtracted images showed a distance between the last coil and a healthy arterial branch too short to add additional coils. Embolization was thus performed using Onyx^®^ (0.4 mL) to fill the afferent artery within the pre-implanted coils and resulted in an immediate complete occlusion, maintained after 13 months follow-up. No leak in the aneurysm or in the vein was reported. A leak upstream the coils in the segmental artery was reported without any consequence on lung perfusion (empty arrowhead). (**A**). Opacification of the afferent artery of a PAVM showed a recanalization through the pre-implanted coils (arrowhead). (**B**). Opacity in the coils and afferent artery showed the distribution of Onyx^®^, with an upstream leak in a segmental arterial branch (empty arrowhead). (**C**). Opacification of the afferent artery showed the absence of opacification of the aneurysmal sac and the efferent vein in favor of immediate occlusion. The opacification of the healthy arterial branch did not reveal any perfusion defect.

**Figure 4 jpm-12-01091-f004:**
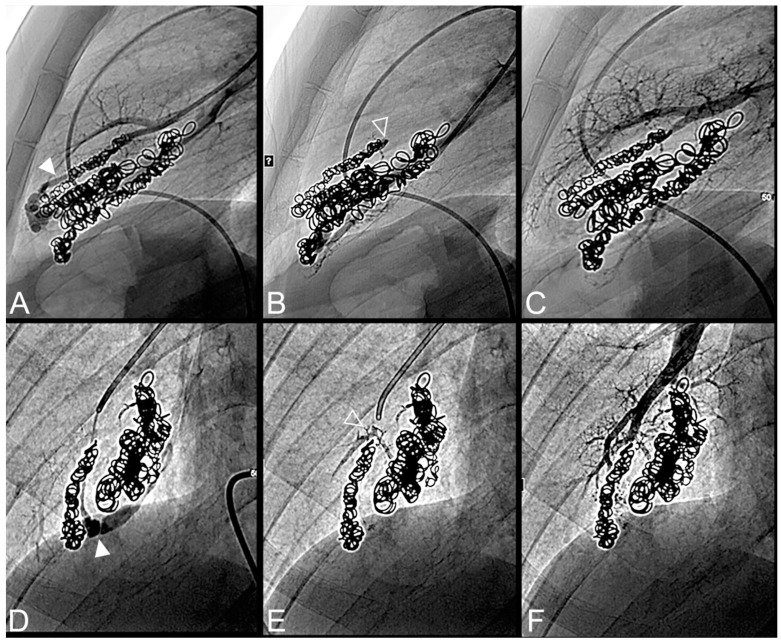
Case example of an 18-year-old man treated for a complex recurrent pulmonary arteriovenous malformation in the middle lobe. Digital subtraction angiography unsubtracted images showed a recanalization in two different segmental feeder arteries (**A**–**F**). Embolization was performed using Onyx^®^ (0.5 mL in each artery) to fill the afferent artery upstream and within the pre-implanted coiling and resulted in an immediate complete occlusion, maintained after 43 months follow-up. No leak in the aneurysm or in the vein was reported. (**A**). Opacification of an afferent artery (full head arrow) showed a recanalization through the pre-implanted coils. (**B**). Opacity upstream and in the last coil (empty arrowhead) showed the distribution of Onyx^®^ without any evidence of a leak in the aneurysmal sac or proximal arterial branch. (**C**). Opacification of the afferent artery showed the absence of opacification of the aneurysmal sac and the efferent vein in favor of immediate occlusion. The opacification of the healthy arterial branch did not reveal any perfusion defect. (**D**). Opacification of a second afferent (full head arrow) artery showed a recanalization through the pre-implanted coils. (**E**). Opacity upstream of the coils showed a leak of Onyx^®^ (empty arrowhead) without evidence of any leak in the aneurysmal sac. (**F**). Opacification of the afferent artery showed the absence of opacification of the aneurysmal sac and the efferent vein in favor of immediate occlusion. The opacification of the healthy arterial branch did not reveal any lung perfusion defect.

**Figure 5 jpm-12-01091-f005:**
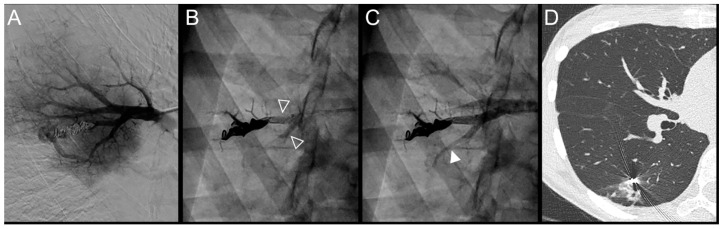
Case example of a 37-year-old woman treated for a simple recurrent pulmonary arteriovenous malformation in the lower right lobe. Embolization was performed using Onyx^®^ (0.5 mL) to fill the afferent artery within the pre-implanted coils and resulted in an immediate complete occlusion, maintained at 34 months follow-up. No leak in the aneurysm or in the vein was reported, but a leak in the upstream sub-segmental arteries was identified. (**A**). Opacification of the afferent artery showed a recanalization through existing coiling. (**B**). Opacity within and upstream of the pre-implanted coils (empty arrowheads) showed the distribution of Onyx^®^, with a leak in a proximal arterial branch. (**C**). Opacification of the afferent artery showed the absence of opacification of the aneurysmal sac and the efferent vein in favor of immediate occlusion. An altered opacification in the upstream branch (full arrowhead) was identified due to the leak of Onyx^®^. (**D**). The one-year follow-up chest CT showed a distal lung infarction related to embolization. Of note, the patient did not suffer from chest pain or pleural effusion after the embolization procedure.

**Figure 6 jpm-12-01091-f006:**
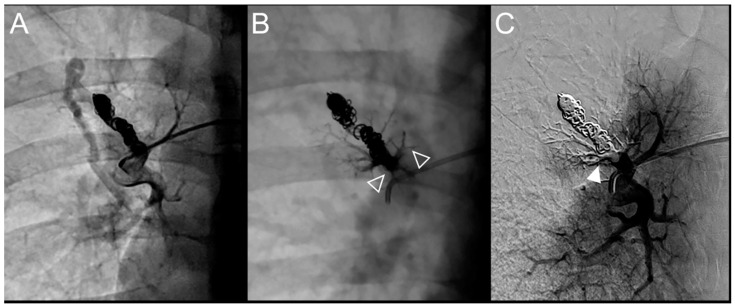
Case example of a 35-year-old man treated for a recurrent simple pulmonary arteriovenous malformation in the right lower lobe. The pulmonary angiograph showed a distance between the last coil and a healthy arterial branch too short to add additional coils. Embolization was thus performed using Onyx^®^ (0.4 mL) to fill the afferent artery upstream and in the pre-implanted coils, without any leak neither in the aneurysm nor in the vein. It resulted in an immediate complete occlusion until 46 months after the procedure when a recurrence was reported. (**A**). Opacification of the afferent artery of a PAVM in the lower right lobe showing a recanalization through the pre-implanted coils. (**B**). Opacity in and upstream (empty arrowheads) the coils showing the distribution of the Onyx^®^, without any evidence of a leak in the aneurysmal sac, but with a leak in the small arterial branches. (**C**). Opacification of the afferent artery showing the absence of opacification of the aneurysmal sac and the efferent vein in favor of immediate occlusion. A perfusion defect was identified in a sub-segmental territory (full head arrow), not related to a symptomatic lung infarction.

**Table 1 jpm-12-01091-t001:** Characteristics of patients.

Population		45
Women		23 (51)
Mean age (SD) (years)		53 (18)
Mean BMI (SD)		25.8 (6.2)
Diabetes mellitus		4 (9)
Tobacco		18 (40)
Mean oxygen saturation (range)		96.3 (92–100)
Curacao criteria		
	Family history of HHT symptoms	45
	Epistaxis	43 (96)
	Telangiectasia	41 (91)
	Liver AVM	7 (16)
	Gastro-intestinal AVM	5 (11)
	Brain AVM	3 (7)
HHT severe complications		
	Hemoptysis	3 (7)
	Brain abscess	5 (11)
	Stroke	7 (16)
Mutation		
	HHT1/ENG	36 (80)
	HHT2/ALK1	5 (11)
	SMAD4	0 (0)
	Unknown/unconfirmed	4 (9)
Unique PAVM		18 (40)
Multiple PAVM		27 (60)

PAVM—pulmonary arteriovenous malformation; HHT—hereditary hemorrhagic telangiectasia; ENG—endogline; SD—standard deviation. Unless otherwise indicated, data are numbers of patients, with percentages in parentheses.

**Table 2 jpm-12-01091-t002:** Pulmonary arteriovenous malformation data before embolization in the overall population and in the two groups (short- and long-term occlusion).

Criteria		Total	Long-Term Occlusion	Short-Term Occlusion	*p*
Previous procedures		70 (100)	42 (60)	28 (40)	NA
PAVMs naive from Onyx^®^		64 (91)	40 (95)	24 (86)	0.166
PAVMs previously treated with Onyx^®^		6 (9)	2 (5)	4 (14)	0.166
Simple PAVMs		61 (87)	36 (86)	25 (89)	0.664
Complex PAVMs		9 (14)	6 (15)	3 (13)	0.664
Mean number of embolizations before onyx per PAVM (SD)		2.0 (1.1)	1.8 (1.1)	2.5 (1.3)	0.01
Mean number of recurrence before Onyx^®^ per PAVM		0.9 (1.1)	0.7 (1.0)	1.4 (1.2)	<0.01
PAVMs first treated with coils		70 (100)	42 (100)	28 (100)	1.00
PAVMs first treated with plugs and coils		2 (3)	2 (5)	0	0.245
Length between aneurysm and plug/coil < 10 mm		43 (61)	27 (64)	16 (57)	0.550
Vein diameter (mm)		4.2 (3.7)	3.4 (1.0)	5.5 (5.6)	<0.01
Aneurysm diameter (mm)		5.1 (5.4)	3.4 (4.1)	8 (6.1)	<0.01
Lobar location					
	Upper right lobe	13 (19)	9 (21)	4 (14)	0.455
	Middle lobe	7 (10)	4 (10)	3 (11)	0.872
	Lower right lobe	22 (31)	11 (26)	11 (39)	0.251
	Upper left lobe	6 (9)	5 (12)	1 (4)	0.226
	Lower left lobe	22 (31)	13 (31)	9 (32)	0.917
Absence of safety distance		33 (47)	17 (60)	16 (43)	0.174
Mechanism of recurrence					
	Recanalization	69 (98)	41 (98)	28 (100)	0.414
	Reperfusion	10 (14)	7 (17)	3 (11)	0.489
	Both	9 (13)	6 (67)	3 (23)	0.664
	Incomplete primary treatment	0	0	0	1.000
Territory potentially at risk					
	Lobar	2 (3)	1 (2)	1 (4)	0.771
	Segmental	18 (26)	9 (21)	9 (32)	0.318
	Sub-segmental	50 (71)	32 (76)	18 (64)	0.284

PAVM—pulmonary arteriovenous malformation. Unless otherwise indicated, data are numbers of initially treated PAVMs with percentages in parentheses.

**Table 3 jpm-12-01091-t003:** Onyx^®^ embolization data and immediate complications in all PAVMs and in the two groups, short- and long-term occlusion.

Immediate Embolization Characteristics		Total	Long-Term Occlusion	Short-Term Occlusion	*p*
Per-embolization occlusion		70 (100)	42 (60)	28 (40)	1.00
Treatment type					
	Onyx^®^ only	53 (76)	31 (74)	22 (79)	0.65
	Onyx^®^ + coils	17(24)	11 (26)	6 (21)	0.65
Onyx^®^ volume, mL (SD)		0.6 (0.5)	0.5 (0.3)	0.7 (0.6)	0.23
Onyx^®^ distribution **					
	Upstream	48 (69)	30 (71)	18 (64)	0.53
	Inside coiling	51 (73)	28 (67)	23 (82)	0.16
	Downstream	13 (19)	7 (17)	6 (21)	0.62
	Upstream + inside	29 (41)	16 (38)	13 (46)	0.49
	Downstream + inside	13 (19)	7 (17)	6 (21)	0.62
	Upstream + inside + downstream	5 (7)	2 (5)	3 (11)	0.35
Upstream leak outside the target					
	At a lobar level	0	0	0	1.00
	At a segmental level	4 (6)	2 (5)	2 (7)	1.00
	At a sub-segmental level	39 (56)	25 (60)	14 (50)	0.68
Perfusion defect in healthy territory					
	Lobar territory	0	0	0	1.00
	Segmental territory	0	0	0	1.00
	Sub-segmental territory	20 (29)	13 (31)	7 (25)	0.59
Downstream leak in draining vein or aneurysmal sac		0	0	0	1.00
Downstream leak in systemic circulation		0	0	0	1.00
Procedure time, min (SD)		105 (34)	110 (33)	99 (35)	0.15
Volume of contrast agent, mL (SD)		110 (49)	116 (50)	102 (47)	0.19
Adverse events during hospitalization time		5 (7.1)	2 (2.8)	3 (4.3)	0.35
	Allergy	1 (20)	0	1 (33.3)	0.22
	Chest pain	1 (20)	0	1 (33.3)	0.22
	Pleural effusion	0	0	0	1.00
	Lung distal infarction	3 (60)	2 (100)	1 (33.3)	0.81
	Lung infection	0	0	0	1.00
	Brain abscess	0	0	0	1.00
	Stroke	0	0	0	1.00
	Access site complication	0	0	0	1.00

** Onyx distribution was characterized according to its presence within the pre-implanted embolic materials and/or downstream and/or upstream of it. Unless otherwise indicated, data are numbers of pulmonary arteriovenous malformations, with percentages in parentheses.

**Table 4 jpm-12-01091-t004:** PAVM follow-up data in the overall population and in the two groups, short- or long-term occlusion.

Long Time Follow-Up		Total	Long-Term Occlusion	Short-Term Occlusion	*p*
Number of PAVMs		70 (100)	42 (60)	28 (40)	
Follow-up time, months (SD)		34.8 (15.3)	25.2 (13.0)	34.1 (16.7)	0.5
Tobacco consumption		28 (40)	14 (33.3)	14 (50)	0.111
Pack-year of tobacco (SD)		9.3 (17.3)	8.8 (2.8)	11.8 (3.1)	0.133
PAVM characteristics					
	Aneurysm diameter, mm (SD)	3.9 (4.7)	2.1 (3.2)	7.0 (5.1)	<0.001
	Vein diameter, mm (SD)	2.9 (1.2)	2.4 (0.9)	3.8 (1.0)	<0.001
	Difference in aneurysm diameter, mm (SD) *	22.9 (37.2)	40.5 (39.1)	4.4 (24.4)	<0.01
	Difference in vein diameter, mm (SD) *	18.7 (21.7)	29.9 (18.2)	0.04 (12.1)	<0.001
Complications					
	Brain abscess	0	0	0	
	Stroke	0	0	0	
	Hemoptysis	2	0	2	
	Hemothorax	0	0	0	

PAVM—pulmonary arteriovenous malformation; SD—standard deviation. Unless otherwise indicated, data are numbers of patients, with percentages in parentheses. * Difference in size was calculated as a proportion of size reduction between baseline and follow-up CT.

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
