# Peer review of "Embolization of Recurrent Pulmonary Arteriovenous Malformations by Ethylene Vinyl Alcohol Copolymer (Onyx®) in Hereditary Hemorrhagic Telangiectasia: Safety and Efficacy"

_jpm, 2022, doi:10.3390/jpm12071091_

Round 1

Reviewer 1 Report

Si-Mohamed et al report the evaluation of safety and efficacy of embolization with a liquid embolic agent for treatment of recurrent PAVMs in HHT. Both short- and long-term results are reported.

The study is well-designed.  The results are reported clearly and are of interest. Some clarifications are needed as far as concerns the characteristics of the study cohort and the definition of outcomes.

However, up to 25% of successful embolizations require second treatment due to PAVM recurrence.

Authors may specify that PAVM recurrence could occur due to potentially different mechanisms, such as true recanalization of a succesfully occluded PAVMs, or growth of a short collateral not amenable treatment at the time of the first embolization.

In the abstracts, Authors state that 45 consecutive patients (51% women, mean (SD) age 53 (18)-year-old) with HHT were retrospectively included in the study, in the Method section. Table 1 and Figure 1 show data regarding 45 pts. However, in the main text, I see no statement regarding the cohort size of the included pts (I assume 45 pts), but only to the number of treated recurrent PAVMs (70). The main text should include the number of enrolled pts, either in the method or in the results section.

Actually, based on Figure 1 flow-chart, the study population is not clear. Authors should clarify Figure 1 as follows: “selected patients= 55” should be changed to “patients with an embolotherapy with Onyx for PAVM recurrence”= 55”. Furthermore, the following box should state “45 pts with with an embolotherapy with Onyx for PAVM recurrence and minimal-1-year- follow-up were included”

Moreover, the following boxes report that 27 patients had suspicion of PAVM recurrence at unenhanced CT, corresponding to 42 PAVMs. Of these 42 PAVMs, 14 of them showed no evidence of recurrence. Therefore, 42 PAVM were defined with persistent occlusion and 28 PAVMs were considered as recurrent lesions.

If I understand correctly, these numbers refer to short-term occlusion (STO) and long-term occlusion (LTO) after treatment with Onyx of PAVM which had already evidence of recurrence before Onyx. If this is true, Authors should change wording, as they are using STO and LTO to indicate the outcome after Onyx re-treatment in the results section, but they are using “evidence of recurrence” to indicate the same outcome in Figure 1, which is confusing. STO and LTO should be defined clearly in the methods, and then used consisently in the flow-chart. 

55 (86%) PAVMs were simple, similarly distributed in the short- and long-term occlusion groups (STO and LTO groups, respectively).

The definition of STO and LTO (as outcome after Onyx re-treatment?) should be given in the methods before, not directly in the results section.

Table1.

HHT2/AKL1

The gene underlying HHT2 is named ALK1, or ACVRL1, not AKL1

Table 2. Pulmonary arteriovenous malformation data before embolization in the overall population and in the two groups (short- and long-term occlusion).

Footnote. PAVM: pulmonary arteriovenous malformation Unless otherwise indicated, data are numbers of patients, with percentages in parentheses.

Actually, numbers seem to indicate the number of performed procedures (i.e. initially treated PAVMs), rather than numbers of patients

Author Response

Reviewer 1

Comments and Suggestions for Authors

Thank you for giving me a chance to review this paper.

The topic of this manuscript is interesting.

I feel this manuscript has a potential value for readers involved in this field.

However, I think that the paper has several issues as below may be addressed. 

My specific comments are followings.

Answer: We thank the reviewer for her/his comment on the present work.

Abstract; OK

Introduction; OK

Methods;

p.3 l.82; Author described that ’70 PAVM were analyzed.” in the result. However, this should be described in the study population. 

Answer: We thank the reviewer for this. We have added this information to the study population section.

p.3 l.101-104; The definition of the absence of safety distance should be described in detail. The expression of 'too short' is ambiguous. How many centimeters is it?

Answer: We thank the reviewer for giving the opportunity to address this question.

We have extended the definition of a safety distance so it now reads: “the radiologists recorded the absence of safety distance defined as a too short distance for coil/plug deployment, i.e., between 0.5 to 1 cm, between the proximal extremity of the primary embolic material used and a healthy upstream artery branch”.

Results;

p.5 l.142 6 PAVM treated a second time should be excluded from this study. I want to know a simple result after the first embolization of Onyx.

Answer: We thank the reviewer for this comment. After excluding PAVM treated twice with Onyx, the rate of success was 56.2% vs 43.8%. We have added this result to the manuscript.

But we kept the former results with inclusion of 70 PAVM because of a long-term recanalization of these PAVM, occurring after a mean (SD) period of 1.9 (0.7) years.

We think that keeping these data is reflecting more the challenge of PAVM embolization in HHT in our clinical practice. We would like to ask kindly the reviewer for her/his understanding.

p.6 l.168; Is PAVM with persistent occlusion in a STO group? I feel the meaning is the opposite. I think PAVM with persistent occlusion is the long-term occlusion.

Answer: We thank the reviewer for pointing out this important point. We agreed that PAVM with persistent occlusion consist in the long-term occlusion group and have edited accordingly it in the whole manuscript. In addition, accordingly to the second reviewer, we have added these terms in the flow chart for a better understanding of the population.

p.7 Table2; please add units of aneurysmal diameter and vein diameter in table 2.

Answer: Corrected.

p.8 Table3; I can’t understand the Onyx distribution. It was written in a very esoteric way.

Answer: We are sorry for the confusion. We have detailed in the footnote the wordings so it now reads: Footnote. “Onyx distribution was characterized according to its presence within the pre-implanted embolic materials and/or downstream and/or upstream of it.”

p.9 l.215; Can the contrast-enhanced CT evaluate the recurrence in the presence of metal artifacts in cases of previous coils?

Answer: We thank the reviewer for this question. Onyx opacity is hampering the visualization of the surrounding tissue such for coils and plugs, mainly to strong beam-hardening like artifacts. This is one the reason for evaluating the vein diameter which is frequently safe of strong artifacts, either on non-enhanced of enhance CT. In our clinical practice, we are performing non-contrast CT. This choice was based on both the current practice in our expert center and on previous results showing a sensitivity for recurrence of 98.4% for PAVMs with a vein diameter larger than 2.5 mm [8].

p9. l.217; This result is unbelievable. You should discuss why this occur.

Answer: We thank the reviewer for giving the opportunity to address this question.

Patients treated in this study underwent multiple embolization before use of Onyx. We think that this may have impacted the vascular compliance of PAVM and result in a retractile and non-compliant PAVM components. But our study did not address this question which opens to further research.

We have then added to the discussion so it now reads: “Finally, follow-up of some PAVMs showed no reduction in vein diameter or aneurysm size despite persistent occlusion, which raises the question of the expected reduction in PAVM size (REF). In our practice, we hypothesized this by a loss in vascular compliance after iterative embolization, opening to furthermore investigations.”

Discussion;

p.15 l.361; The author’s method of Onyx injection was focused to avoid the downstream leakages. However, the present basis of embolization for PAVM is to embolize both nidus and feeding arteries. Rather, it would be better to embolize the downstream artery and nidus by Onyx. Please refer to the following papers; 1) Eur Radiol. 2021 Jul;31(7):5409-5420. 2) J Vasc Interv Radiol. 2021 Jul;32(7):993-1001.

Answer: We thank the reviewer for pointing out these papers that we added in the discussion. We have then edited the discussion so it now reads: “Despite the evidence for treating the nidus in addition to the feeding artery in PAVM naïve of embolization (REF), we avoided downstream leakages by stopping the procedure when Onyx® would go past the materials, which occurred in 19 % of cases.”

REF: 1) Eur Radiol. 2021 Jul;31(7):5409-5420. 2) J Vasc Interv Radiol. 2021 Jul;32(7):993-1001.

p.15 l.366; The upstream leakages were reported in 69% cases. This fact may indicate the difficulty of control of Onyx. Since there were no major complications of Onyx for PAVM, it is premature to describe that there was no problem with upstream leakages. To avoid upstream leakages, NBCA injection under flow control may be better. 

Answer: We thank the reviewer for this comment. We agree on the possibility to inject Onyx under flow control.

We have detailed the discussion so it now reads: “The procedure was also stopped when Onyx® would go upstream the pre-implanted embolic material in a healthy arterial branch. Nevertheless, in 69% of cases, an upstream leak in a non-involved arterial branch was reported which opens to injection techniques under flow control (REF).”

REF:  Shi, Zhong-Song et al. « Flow Control Techniques for Onyx Embolization of Intracranial Dural Arteriovenous Fistulae ». Journal of Neurointerventional Surgery 5, no 4 (juillet 2013): 311‑16. https://doi.org/10.1136/neurintsurg-2012-010303.

p.16 l.377; 'This complication would have been more frequent using coils or plugs.' What evidence does this have? If there is no reference, please delete this sentence.

Answer: We thank the reviewer for this relevant point. We have added 1 reference supporting a rate of lung infarction at 59% in a PAVM population using coils.

Ref: Brillet P-Y, Dumont P, Bouaziz N, et al (2007) Pulmonary arteriovenous malformation treated with embolotherapy: systemic collateral supply at multidetector CT angiography after 2-20-year follow-up. Radiology 242:267–276. https://doi.org/10.1148/radiol.2421041571

Reviewer 2 Report

Thank you for giving me a chance to review this paper. 
The topic of this manuscript is interesting.
I feel this manuscript has a potential value for readers involved in this field.
However, I think that the paper has several issues as below may be addressed. 
My specific comments are followings.

Abstract; OK

Introduction; OK

Methods;

p.3 l.82; Author described that ’70 PAVM were analyzed.” in the result. However, this should be described in the this study population. 

p.3 l.101-104; The definition of the absence of safety distance should be described in detail. The expression of 'too short' is ambiguous. How many centimeters is it?

Results;

p.5 l.142 6 PAVM treated a second time should be excluded from this study. I want to know a simple result after the first embolization of Onyx.

p.6 l.168; Is PAVM with persistent occlusion in a STO group? I feel the meaning is the opposite. I think PAVM with persistent occlusion is the long-term occlusion.

p.7 Table2; please add units of aneurysmal diameter and vein diameter in table 2.

p.8 Table3; I can’t understand the Onyx distribution. It was written in a very esoteric way.

p.9 l.215; Can the contrast-enhanced CT evaluate the recurrence in the presence of metal artifacts in cases of previous coils?

p9. l.217; This result is unbelievable. You should discuss why this occur.

Discussion;

p.15 l.361; The author’s method of Onyx injection was focused to avoid the downstream leakages. However, the present basis of embolization for PAVM is to embolize both nidus and feeding arteries. Rather, it would be better to embolize the downstream artery and nidus by Onyx. Please refer to the following papers; 1) Eur Radiol. 2021 Jul;31(7):5409-5420. 2) J Vasc Interv Radiol. 2021 Jul;32(7):993-1001.

p.15 l.366; The upstream leakages were reported in 69% cases. This fact may indicate the difficulty of control of Onyx. Since there were no major complications of Onyx for PAVM, it is premature to describe that there was no problem with upstream leakages. To avoid upstream leakages, NBCA injection under flow control may be better. 

p.16 l.377; 'This complication would have been more frequent using coils or plugs.' What evidence does this have? If there is no reference, please delete this sentence.

Author Response

Si-Mohamed et al report the evaluation of safety and efficacy of embolization with a liquid embolic agent for treatment of recurrent PAVMs in HHT. Both short- and long-term results are reported. The study is well-designed.  The results are reported clearly and are of interest. Some clarifications are needed as far as concerns the characteristics of the study cohort and the definition of outcomes. However, up to 25% of successful embolizations require second treatment due to PAVM recurrence. Authors may specify that PAVM recurrence could occur due to potentially different mechanisms, such as true recanalization of a succesfully occluded PAVMs, or growth of a short collateral not amenable treatment at the time of the first embolization.

Answer: We thank the reviewer for this comment.

In the abstracts, Authors state that 45 consecutive patients (51% women, mean (SD) age 53 (18)-year-old) with HHT were retrospectively included in the study, in the Method section. Table 1 and Figure 1 show data regarding 45 pts. However, in the main text, I see no statement regarding the cohort size of the included pts (I assume 45 pts), but only to the number of treated recurrent PAVMs (70). The main text should include the number of enrolled pts, either in the method or in the results section.

Answer: We thank the reviewer for this comment. We have added this information in the methods and results.

Actually, based on Figure 1 flow-chart, the study population is not clear. Authors should clarify Figure 1 as follows: “selected patients= 55” should be changed to “patients with an embolotherapy with Onyx for PAVM recurrence”= 55”. Furthermore, the following box should state “45 pts with with an embolotherapy with Onyx for PAVM recurrence and minimal-1-year- follow-up were included”

Moreover, the following boxes report that 27 patients had suspicion of PAVM recurrence at unenhanced CT, corresponding to 42 PAVMs. Of these 42 PAVMs, 14 of them showed no evidence of recurrence. Therefore, 42 PAVM were defined with persistent occlusion and 28 PAVMs were considered as recurrent lesions.

If I understand correctly, these numbers refer to short-term occlusion (STO) and long-term occlusion (LTO) after treatment with Onyx of PAVM which had already evidence of recurrence before Onyx. If this is true, Authors should change wording, as they are using STO and LTO to indicate the outcome after Onyx re-treatment in the results section, but they are using “evidence of recurrence” to indicate the same outcome in Figure 1, which is confusing. STO and LTO should be defined clearly in the methods, and then used consistently in the flow-chart. 

Answer: We thank the reviewer for her/his comments. We have clarified the flow chart accordingly.

55 (86%) PAVMs were simple, similarly distributed in the short- and long-term occlusion groups (STO and LTO groups, respectively).

The definition of STO and LTO (as outcome after Onyx re-treatment?) should be given in the methods before, not directly in the results section.

Answer: We thank the reviewer for this comment. We have edited the methods so it now reads: This allows to define two groups, i.e., a long-term occlusion (LTO) group for patient with persistent occlusion at follow-up, and a short-term occlusion (STO) group for patient with occlusion immediately after embolization but with recurrence at follow-up.

Table1. HHT2/AKL1. The gene underlying HHT2 is named ALK1, or ACVRL1, not AKL1.

Answer: I thank the reviewer for this remark. It is indeed a typing error. This point has been changed.

Table 2. Pulmonary arteriovenous malformation data before embolization in the overall population and in the two groups (short- and long-term occlusion). Footnote. PAVM: pulmonary arteriovenous malformation Unless otherwise indicated, data are numbers of patients, with percentages in parentheses. Actually, numbers seem to indicate the number of performed procedures (i.e. initially treated PAVMs), rather than numbers of patients

Answer: We thank the reviewer for this comment. We are sorry for the confusion. We have edited accordingly the footnote.
